# Nature-Based Mindfulness: A Qualitative Study of the Experience of Support for Self-Regulation

**DOI:** 10.3390/healthcare11060905

**Published:** 2023-03-21

**Authors:** Dorthe Djernis, Cecilie M. Lundsgaard, Helle Rønn-Smidt, Jesper Dahlgaard

**Affiliations:** 1Department of Geosciences and Natural Resource Management, University of Copenhagen, 1958 Frederiksberg, Denmark; 2Department of Psychology, University of Copenhagen, 1353 Copenhagen, Denmark; 3Program for Mind and Body in Mental Health, Research Center for Health and Welfare Technology, VIA University College, 8200 Aarhus, Denmark

**Keywords:** nature-based, mindfulness, meditation, natural environment, self-regulation, IPA

## Abstract

Self-regulation is pivotal for human well-being and mental health. In this qualitative study, we followed a randomized controlled trial (RCT) and explored how a five-day residential mindfulness program in a restorative natural setting supported self-regulation among university students experiencing moderate to severe stress. Six participants were interviewed post intervention and at three months’ follow-up on how they experienced the retreat. Through interpretative phenomenological analysis, four interrelated themes emerged: “supportive conditions”, “attitudes of mindfulness”, “connection” and “physical and psychological balance.” These themes reflected the outcomes of the retreat that participants valued in stressful situations. A progression occurred during the retreat through the themes, with emphasis developing from the supportive conditions of the setting, to cultivating mindful attitudes, over connection to both self, others and nature, to changes and effects on the physical, psychological and even spiritual level. In addition, participants emphasized experiences of positive emotions, energy, calmness, meta-awareness and the feeling of being part of the web of life. In conclusion, participants’ experiences with a five-day nature-based mindfulness intervention revealed a range of qualities of both physical-, psychological-, social- and spiritual nature that are supportive for self-regulation.

## 1. Introduction

Staying in nature may be associated with mental health qualities, such as reduction in symptoms of stress, depression and anxiety [1,2] and promotion of attention, emotion regulation and positive emotions [3,4]. For healthcare to benefit from these qualities, a variety of nature-based interventions were developed, including green care and ecotherapy (for an overview, see [5]). Over the last decade, a growing interest emerged for incorporating mindfulness into nature-based interventions, since mindfulness also promotes awareness and attention, self- and emotion-regulation, and reduces symptoms of depression and perceived stress. Examples include forest bathing, wilderness therapy [6,7,8] and interventions with nature and mindfulness for students [9,10] and healthy workplaces [11]. However, this field is still in its infancy [12].

Mindfulness was defined as “paying attention in a particular way: on purpose, in the present moment, and non-judgmentally” [13] (p. 4). It is an innate capacity that can be trained, e.g., through the mindfulness-based stress reduction program (MBSR) created by Jon Kabat-Zinn [14]. The health-promoting effects of mindfulness-based interventions (MBIs) were extensively documented in recent decades and include reduction in symptoms of stress, anxiety and depression [15,16]. Attention and self-compassion were shown as key mechanisms from studies on mindfulness-based interventions (MBIs) and they allowed and supported self-regulation and mental health [17,18]. Most studies of meditation were based on programs with weekly sessions, but retreats also showed positive effects on, e.g., symptoms of depression, anxiety, stress and on quality of life [19].

In recent years, interest developed to include mindfulness in nature-based interventions and vice versa [20] to improve efficacy. A recent meta-analysis comparing mindfulness-based interventions indoors versus in nature revealed that nature-based mindfulness was moderately superior to mindfulness conducted in non-natural settings [12]. Furthermore, nature-based mindfulness interventions (NBMIs) revealed a medium size effect (*g* = 0.54) when combining psychological, physiological and interpersonal outcome variables [12]. One of the included studies that examined manual-based mindfulness training in nature showed positive effects on awareness [20] of a six-week mindfulness class in a natural setting. Furthermore, qualitative studies found that meditation supports the experience of connection to nature [21], and that attitudes of mindfulness promotes awareness of the natural environment [22]. More studies found that combining mindfulness training with nature-based activities integrates the qualities of being mindful with connecting to nature, and improves measures of mental health [23,24]. Among the few mixed-methods studies in the field of NBMIs, one study on mindful walks in nature showed positive changes in mindfulness [25] and affect [26]. Mindfulness training and experiencing nature was documented to affect self-compassion and awareness, which both support self-regulation [9,12]. The added effect of training mindfulness in natural settings may stem from participants more easily connecting with sensory impressions improving their experience of connectedness to nature [9] and allowing them to benefit more from the restorative processes offered by the natural environments. Additionally, the enhanced awareness of seeking restorative environments may explain positive results in follow-up studies [27].

Self-regulation (SR) is pivotal for well-being and mental health [28] and the question how nature-based mindfulness supports self-regulation is the focus of this qualitative study. SR is referred to as the psychological process that unfold with a purpose of regulating emotions, thoughts and actions in a desired and intended way [17,29]. Self-regulation also is referred to as a quality or part of attention and awareness, prone to depletion just like attention and awareness per se [29]. Both meditation and staying in nature were shown to be sources of strengthening an individual’s potential for self-regulation and executive functioning [29] through enhanced attention [17], with meditation supporting attentional, emotional and behavioral self-regulation [30].

The five-day NBMI retreat presented in this article also was evaluated by quantitative methods [9]. The findings of that evaluation were improved self-compassion, attention, awareness, and a buffering of increasing stress levels during the semester, for both an indoor and an outdoor mindfulness retreat setting. To gain wider knowledge on how this NBMI was experienced, and to triangulate the quantitative findings [31], interviews with participants were analyzed using interpretative phenomenological analysis (IPA). IPA is a method inspired by phenomenology, hermeneutics and ideography, and the method was shown to be well suited to produce rich data on participants’ personal lived experiences [32,33,34].

The aim of the present study, thus, is to qualitatively investigate whether and how the NBMI retreat affected the participants’ experience of support for self-regulation, with specific emphasis on the following:

How did participants in a manual-based five-day residential mindfulness retreat experience mindfulness training, nature and the group as supporting self-regulation, and what sense did they make of this?

## 2. Materials and Methods

### 2.1. Participants

University students with self-reported perceived stress (PSS ≥ 16) were recruited by means of intranet announcements, flyers and referrals by university counselors. The criteria for participating in the trial were as follows:Participants were active bachelor’s or master’s degree students at Danish universities or University Colleges;Participants experienced elevated stress at the time of enrolment, indicated by a PSS score of 16 or above;No known psychiatric diagnosis such as severe depression, severe anxiety, adjustment disorder, post-traumatic stress disorder, personality disorder or psychosis, and no known autism or untreated attention deficit hyperactivity disorder;No self-reported risk of suicide or addiction to alcohol, tobacco or drugs.

After screening and visitation interviews, the students were randomized into three groups: a waiting list control group, an indoor-based mindfulness group and an NBMI group. Seven participants of the NBMI group were invited for interview, and they all accepted. One interview was excluded for technical reasons, however. The group reflected the main sample: five of the six were women, all were of Danish ancestry, and their mean age was 28 years. The study was approved by the National Committee on Health Research Ethics (protocol 51986), the Danish Data Protection Agency (protocol 2015-57-0116) and preregistered at ClinicalTrials.gov (NCT02867657). No adverse effects were expected or experienced.

### 2.2. Intervention

The retreat program was based on the eight-week MBSR program, condensed into a five-day retreat that included all the MBSR sessions and informal mindfulness between sessions, with added walking and sitting meditations as well as yoga in the mornings and evenings. On day 4, the participants were silent for 24 h, possibly enabling an enhanced focus on their own practice and inner experiences. The program was described briefly in Djernis et al. [9] and is available from the corresponding author upon request.

Participants were outdoor during the entire retreat for all activities including practicing mindfulness, breaks, meals and when sleeping in solo tents (Figure 1). The setting was a 3.5-acre forest garden designed for NBMIs and situated in an arboretum. The weather during this retreat was mostly sunny, with temperatures ranging from 50 to 86 degrees Fahrenheit/10 to 30 degrees Celsius. The retreat format was chosen to make transportation to this restorative countryside environment more feasible for the students, who mostly lived in cities.

### 2.3. Interviews

At the end of the NBMI retreat, in 2016, three interviews were conducted in the therapy garden. Another three participants were interviewed online a few days after the retreat. The same six participants were interviewed online at three months’ follow-up. The interviews were individual, semi-structured and open-ended, and lasted between 20 and 43 min. To support their memory, participants had the option to refer to the logbooks they were given to complete during the retreat. To gain knowledge of participants’ sense-making regarding how the program influenced their experiences of their own self-regulation unfolding, the interview guide focused on challenging real-life situations and concrete experiences of support related to those situations. The questions were:

Tuning in:How are you today?How have you been during recent days/since the retreat?Have you experienced any challenging situations during recent days/this period?What was helpful when dealing with these issues? (If this was not mentioned by participants themselves, probing questions were:In your own approach?From other people?From the environment?)

The first author (51 years old, a clinical psychologist of Danish ancestry, and PhD student who participated in the retreat as project manager) conducted the interview. The students were compliant during the interviews; some needed more structure and active participation from the interviewer than others, and the interviews were conducted accordingly. All 12 interviews, whether in person or online, were audio- and video-recorded (using Zoom.us). The video-recording did not seem to attract much attention, which may be explained by the camera being invisible inside the computer. The interviews were transcribed verbatim by a professional transcriber and a student assistant, and all identifying information was removed. In the presentation of the results, quotes from the transcripts are shown with ellipses to indicate pauses in speech, while ellipses in square brackets […] indicate material omitted or added for clarity. The interview guide is available from the first author upon request.

### 2.4. Data Analysis

The approach in this paper was qualitative, using the methodology of IPA to examine the participants’ personal lived experiences in detail [32,33,34,35,36]. IPA can produce rich and illustrative data that highlight novel factors [37]. The theoretical underpinning of IPA is a combination of phenomenology, hermeneutics and ideography [38], allowing the phenomenon to be seen afresh on its own terms, and facilitating the emergence of the novel and unexpected [38]. The aim is to connect the interpretation of the phenomena to a wider social, cultural and theoretical context to answer the research questions [39].

According to the IPA framework, a person is self-reflexive and the expert regarding his/her own engagement with the world [33,34]. The aim of the research is to negotiate a mutual understanding of an experience between researcher and participant, allowing intersubjective meaning-making by both parties [34]. The researcher is also an inclusive part of the world being described [32], and this interpretative engagement with text and transcripts—the researcher being the primary analytical instrument—must be acknowledged [36]. This also implies the obligation for researchers to put any preconceptions aside and strive to see the world as the participant sees it, understanding the person-in-context [32].

IPA was shown to be well suited to examining individual experiences such as trauma, rheumatism and pain [33,36,38,40]. Experiences of mindfulness were also explored using IPA [36,41]. We argue that not only does IPA provide a relevant methodology to study participants’ experiences of mindfulness in nature, but its phenomenological epistemology also correlates strongly with the philosophy upon which mindfulness is based. Both aim to be as close as possible to current experience as it is—lived experience. This means letting experiences reveal themselves as much as possible, without evaluation or condemnation. According to phenomenology, personal experiences have a structure that extends across space and time. Every person is fixed in the present, and the present transcends the present, always including the past and the future [42]. This resembles the focus on the here-and-now in mindfulness.

The first and second author completed the following steps during the process of analysis:All transcripts were read and coded for emotion-laden words or sentences, and emergent patterns (themes) were identified;A “dialogue” between the coded data and our a priori knowledge of psychology and mindfulness was developed through discussion in two rounds;This “dialogue” continued until the meanings and themes seemed clear and were decided upon;Related transcript extracts were grouped for each theme; some could be included within two themes;The relationships between the themes were structured;A narrative was developed based on the structured transcript extracts, and a figure was drawn.

The third author triangulated the analysis by comparing the transcripts with the outcome of the process.

## 3. Results

Four overarching themes emerged:“Supportive conditions” included how participants experienced the setting of the retreat; this theme had the subthemes “having a break” and “sensing nature”;“Attitudes of mindfulness” involved the experience of the central attitudes of the MBSR program;“Connection” included three aspects of experiencing connection: “connection to the group”, “connection to nature” and “connection to oneself”;“Physical and psychological balance” included the subthemes “calm and focus”, “energy and happiness” and “insight”.

All four themes reflected factors valued by participants in stressful situations, and they were interrelated: enhancing one factor affected the others. For instance, finding support in the retreat conditions made it easier to access positive emotions and energy. Another example was that integrating mindfulness attitudes, such as acceptance, enhanced presence and calm, allowed a deeper connection to nature (Figure 2).

### 3.1. Supportive Conditions

Participants described how being on the retreat site, away from their everyday lives, and focusing on sensory stimulation affected both their general state and their mindfulness practice positively. Indirectly, this was helpful when they met challenges during the retreat.

#### 3.1.1. Having a Break

From the beginning of the retreat, participants felt that the setting was different from their usually busy lives. It could be hard for them to turn off their cell phones and leave everything behind, but it was also like having a break. Some experienced immediate physical relaxation and the revitalization of breathing fresher air. Most participants felt that they did not have to make decisions or judgments when they were outside in nature, where there were no expectations to meet.

Jane: “When you go outside, you get a distance from all the stimulation you find indoors. Maybe that is why you do not need to concern yourself with it. Then, it is just pure nature, which you in fact do not need to make any judgments about.”

Because participants were new to the setting and to each other, it seemed to make a change of perspective easier.

#### 3.1.2. Sensing Nature

Participants were asked to be mindful between sessions but were not instructed to pay any particular attention to the environment. However, they all experienced nature as stimulating, both during and between sessions, and nature was, therefore, found to support mindfulness practices and present-moment awareness per se. Nature was found to stimulate all the senses:

James: “The soundscape, you can suddenly hear some owls, or you notice the wind moving some leaves or how the stream moves and finds its way. It is just something that really encapsulates the moment and just makes you happy that you are just there in that moment and not really thinking about what is going to happen tomorrow or what has happened before.”

The natural environment could also be distracting and, for example, insects could give rise to disgust. When brought into the meditation, this could be a welcome challenge (Jane: “Well, it just got to be a part of being out here. Maybe it supported the sense of being at one with nature”). The variations in the natural setting seemed important, as they allowed the participants to choose between different natural settings depending on their current needs.

### 3.2. Attitudes of Mindfulness

The participants attended a mindfulness class that included the cultivation of mindful attitudes such as nonjudgment, acceptance, non-striving and letting go. It seemed that they were motivated and able to use the teachings immediately (Olivia: “One of the first things was the thing about nonjudgment—it was so good. Maybe it made it easier to engage and feel more settled”). Not surprisingly, learning points from the class could be seen in their responses when they were asked what had been helpful in critical or stressful situations. Most participants experienced enhanced self-acceptance in relation to both their hidden emotions and their overt behavior. They sensed acceptance from the group, and they welcomed each other, whoever they were:

Silvia: “We are relating to each other, the others are here, we want to relate, and we want the best for each other—but I am freed from thinking whether anything I said might be stupid, and I don’t have to evaluate the others.”

Acceptance seemed to create a space for letting go of critical and judgmental thoughts, and vice versa; and with more kindness toward themselves and each other, a feeling of being okay was prominent. Letting go of judgmental or controlling thoughts was found to be mirrored by the natural environment (Laura: “Nature doesn’t need to control either, it is just there, with conditions as they are”).

Being with one’s sensations (e.g., bodily pain during sitting meditations), emotions and thoughts, whether positive or negative, led to some deeper mind–body processes:

James: “A knot I had inside, which I have never had contact with before, was dissolved in an explosion of different emotions: happiness and sadness and anger and other emotions which slipped away and evolved into a sense of freedom.”

Allowing space for difficult emotions could be challenging, especially when they were intensified by ruminative thinking. Looking at, or sensing, the environment was, at times, used as a diversion, enabling letting-go and a more decentered stance towards unpleasant thoughts.

### 3.3. Connection

Contrary to feeling separated, fragmented or isolated, participants expressed feelings of being connected, being whole and being in a process together. This connectedness to themselves, to others in the group, and to nature was experienced as supportive when they were challenged emotionally.

#### 3.3.1. Connection to the Group

Participants had an enhanced sense of community. They felt that they were engaged in this process together and were somehow on the same wavelength, with trust, acceptance and even love. There was consensus about what was going to happen, and everyone created a nonjudgmental atmosphere together. One described the feeling of community even on the silent day:

James: “You felt a community. Even though you did not see them or spoke to them, there was still this sense of community, because you felt you were doing this together, and you were not alone, and it was okay to feel the feelings that you had.”

The participants described how they created a feeling of safety together that allowed them to share their experiences and to work with themselves on a deeper level.

#### 3.3.2. Connection to Nature

Connection to nature took many forms, such as getting close to an animal, experiencing eye contact and sensing another life intimately; or a deep experience of being nature oneself, like being an animal. These experiences were associated with gratitude and a sense of coming home:

James: “Suddenly, I stared at a leaf and started to cry—because of the leaf, which just lay there by itself, and I found that there was something deeper going on. It seemed that calm and presence allowed one to connect to nature by stepping out of clock time and becoming immersed in one’s experience.”

#### 3.3.3. Connection to Oneself

Participants experienced more connection to themselves than usual. For some, it had been years since they related to themselves in this way. The feeling of wholeness and just being oneself related to the experience of knowing oneself:

Jane: “A totally relaxed feeling where I can have an overview of things, and I can feel what is good and what is bad [for me], and I do not let myself be overwhelmed by the world around me and things I must do.”

Connectedness to oneself had an aspect of being connected to the body, and most participants mentioned grounding and sensing the mind–body connection as a tool for self-regulation, such as sensing one’s feet when emotionally unbalanced. Another aspect was sensing oneself more clearly as part of common humanity (“feeling a little more like a human”).

### 3.4. Physical and Psychological Balance

When participants experienced difficult situations, they drew on what they had learned in the mindfulness training and on the retreat as such, both specifically and generally. This involved knowing how to build a capacity for calm, focus, energy and happiness, and how to access those resources. Emerging insight also seemed to strengthen self-efficacy, thus building the capacity for self-regulation.

#### 3.4.1. Calm and Focus

A few days into the retreat, when the participants were more settled, peace seemed to descend. Some found a calm they had not known for a long time. It was found in nature, in the contrast with their busy lives back home, and especially when they were close to water (the pond or stream) or connecting mindfully to animals and green nature. Another source of calm was found in the group:

Silvia: “Twenty-four hours of silence, together with people you felt safe with, gave me such a sense of peace.”

The sense of stability, which was cultivated in the mindfulness meditations and in relation to the natural environment seemed to be embodied and was something to draw on:

James: “It gives me a sense of stability, I feel more solid, and it makes me go less into my thoughts.”

At follow-up, most participants still found it easier to focus and concentrate in school, work and other life situations.

#### 3.4.2. Energy and Happiness

Energy and happiness seemed to be important when participants were asked what was helpful in stressful situations. The present-moment awareness cultivated in the mindfulness sessions caused positive emotions (Olivia: “It is just happiness, I think, because I am able to be in the present moment”). Supportive emotions also appeared spontaneously and, at times, seemed to occur on a more existential level:

Jane: “It makes me feel blissed out, I get happy. It’s just like being back, I mean all the way back to the starting point, before all the impact from outside, which did all sorts of things. It is just me.”

Sensory impressions experienced during contact with the natural environment also affected emotional states directly: for instance, changing colors, sunshine and the movements of an animal could evoke happiness, peacefulness or relaxation. The feeling that “nature” was generous, without expecting anything in return, seemed to make participants relax and just receive. This, in turn, made room for happiness and gratitude when special moments in nature were experienced. A gain in energy was another positive aspect of being in nature:

Silvia: “I am more aware of how nature affects my mood, and if I am low on energy, it can be energizing to take a walk, or to be in nature and get a bit happier.”

#### 3.4.3. Insight

Insight, from the mindfulness training and from being in nature, seemed pivotal—both in handling everyday stressors and in expanding the scope of life. Most prevalent was the recognition that thoughts can be given too much attention and importance, and that they may neither reflect reality nor be helpful. Being more aware of one’s inner landscape by gaining a distance (i.e., decentering) from one’s thoughts allowed more space and a perspective of meta-awareness:

Laura: “Thoughts are not reality. […] I was really surprised by the difference between a little fear, where I could manage my thoughts, and then when they got overwhelming and dominating.”

Gaining new insight made participants reflect on how they could have taken better care of themselves in the past, and how they would be able to use this insight in the present or future.

Olivia: “I am more aware of myself and able to say no in situations where I am uncomfortable, where in the past I would have gone further and put up with too much.”

At times, these insights about the past were accompanied by sadness, but when participants looked into the future, there seemed to be confidence that they had the capacity to change some habits concerning their thoughts and behaviors. At the three-month follow-up, this confidence still seemed strong.

This supportive insight, at times, had an existential character, as a deep knowing that one needed to be silent to hear the insight from within. The feeling that one was a small and natural part of the web of life also seemed to be a common insight, whether new or previously experienced:

Silvia: “I feel like life—I mean as I see it now, I don’t know how I would respond later—but I think that life is not that complicated, it is much simpler than what we make of it.”

In summary, the findings roughly indicate a progression during the retreat (Figure 3). The setting is central to self-regulation, which affects the capacity for mindfulness, which, in turn, leads to connections on a deeper level, ultimately producing insight.

## 4. Discussion

Participants’ experiences in this NBMI study supported self-regulation in several ways, including the way they related to themselves and in their relations with others and the environment. Practical (e.g., retreat introduction), relational and existential factors influenced the experience of support, and these factors were interrelated. This interrelatedness highlights the importance of understanding and treating NBMIs as complex interventions where each factor may affect all others.

Shapiro and Schwartz [43] discussed the relation between mindfulness and self-regulation. They suggested that both intention and attention enhance self-regulation through improved connection. Being willing to attend to the present moment and to reperceive, even when that which is experienced is uncomfortable (e.g., when the propensity towards reactive behavioral patterns are strong), makes a closer connection possible, that may facilitate self-regulation, ultimately improving health: “Intention → attention → connection → regulation → order → health” [44] (p. 380). The process experienced by the students in this study, to some degree, resembled the process outlined by Shapiro et al. [44]. In addition, nature also may enhance the experience of support towards self-regulation, either by itself or through interactions with other active ingredients, such as mindfulness training and social factors.

### 4.1. Theme 1: Supportive Conditions

When the students arrived, leaving their stressful everyday lives behind to immerse themselves in nature, they were setting the intention for the retreat. It took some effort to acclimatize such as, e.g., turning off their cell phones, but they felt welcomed by nature, the others and attitudes of mindfulness including acceptance. Nature did play a supportive role possibly supported by attention restoration following ART [29]. ART holds that when an environment has the property of making one feel distant from whatever has depleted one’s resources, it is more likely to support the restoration of attention. The self-regulating effect of feeling welcome by nature, the group and the mindful attitudes experienced can, thus, be explained in terms of attention that is restored when exposed to nature [29] and by cultivating mindfulness [44].

### 4.2. Theme 2: Mindfulness Attitudes

Being in the natural environment seemed to support students’ presence and mindfulness attitudes. This is in line with a large body of research showing that attention restoration is positively affected by exposure to nature [2]. As shown by Schuling et al. [26], being mindful in nature draws attention to the senses, which can be experienced as “bathing” in sensory input, enhancing well-being and self-confidence. Steven Kaplan [29] suggested that the process of restoring attention in nature resembles the process of meditation: nature can be so interesting that it holds one’s effortless attention, in a way that may be comparable to the way of relating, with nourishing attitudes of kindness and self-compassion to the flow of sensations and thoughts during meditation. In addition, for the untrained meditators for whom meditation is effortful, it may be supportive to practice in nature, where no training is needed to gain the restorative benefits.

### 4.3. Theme 3: Connection

The sense of connection was present for all students and seemed to be an important ingredient in the process of self-regulation. Experiences of connectedness often emerged in NBMI studies. Riley [21], for instance, showed that the compassion, empathy and surrender experienced on a five-day mindful hike created stronger and deeper experiences of connections to nature. Connections with oneself, with the group and with nature may interact, as found in another study by Sidenius et al. [24]. Here, severely stressed individuals on a 10-week NBMI experienced the natural setting, the people and the mindful activities as an interacting “whole”, provided a basis for the becoming more aware of themselves as “human wholes”. Another study investigated the relationship between connectedness to nature and the experience of the restorative effect of nature and found that people who experienced more connectedness to nature also received more of the beneficial restorative effects from nature exposure [45].

### 4.4. Theme 4: Physical and Psychological Balance

The retreat appeared to bring about psychological balance through self-regulation. Effects of calmness, peace and concentration remained until follow-up and were found to support self-regulation. The same applied to increased positive emotions with experiences of energy and happiness, which were especially gained from the natural environment. Positive emotions may improve self-regulation, affecting both physical and psychological balance. The broaden-and-build theory of positive emotions revealed that positive emotions increase the capacity for emotion regulation [46] and broaden awareness and people’s momentary thought-action repertoires [47]. This allows for new and more flexible thoughts and actions, thus building up personal resources that improve resilience, well-being and health [46]. In addition to increased positive emotions from nature exposure, the inclusion of loving-kindness meditation in the MBSR-curriculum may also have increased positive emotions such as happiness and love contributing to self-regulation [48]. Meta-awareness was enhanced, including the insight that “thoughts are thoughts” and not the truth. Finally, what seemed to be an existential or spiritual insight shared by most participants was the sense of connecting to something bigger than oneself; being part of the web of life.

### 4.5. Strength and Limitations

These findings seemed to qualify the quantitative counterpart of this study [9], not so much by identifying paradoxes, tensions or diversity, but rather by pointing to aspects and deeper levels of support for self-regulation experienced during the NBMI retreat and reported through interviews within a few days following the retreat. These findings may explain the participants’ enhanced levels of mindfulness and self-compassion [9]. Additionally, the results are in line with existing qualitative research; and they addressed areas to be considered during the planning and conduct of NBMI retreats. The purpose of IPA is not to produce generalizations, but rather to give an in-depth description of how participants are affected—in this case, by meditating in nature. Such findings may be transferred to other populations if idiosyncrasy and complexity are considered [1,35].

The fact that the first author participated in the retreat as a project manager may have made the interviewees overly compliant, leading them to omit material they assumed might contradict the outcomes the interviewer wanted. Interviewer biases may also have been present. Not asking about obstacles to self-regulation could be a limitation, as could the omission of interviews with the indoor group. Such interviews might have qualified the findings, but they would also have broadened the scope of the study at the expense of its depth. The fact that three of the six interviews post-intervention were carried out online a few days later might have caused some details to be recalled less vividly by the participants; however, we observed no systematic differences among participant experiences when analyzing the recordings for the IPA. Member-checking was not used in this study, but the third author triangulated the findings that emerged from the process between the first and second authors, i.e., to add perspectives and unify the findings.

The outcome of the intervention in terms of self-regulation, thus, may stem from both the mindfulness training and the natural setting, exemplified by the four identified and interrelated themes. Future research may show how interaction with the natural environment enhances the effects of mindfulness training; for instance, by focusing on the themes found in this study. Different types of mindfulness practice could be investigated, as well as different restorative environments and populations.

## 5. Conclusions

The natural setting of the intervention appeared to support the capacity for self-regulation cultivated in the mindfulness training. The most emphasized qualities were the sense of being away from everyday life and immersed in nature, and the sense of connection with oneself, others and nature. Thus, the opening and connecting to inner experiences, from being in nature, supported the mindfulness skills acquired. The positive outcomes were happiness, energy, calm and meta-awareness. These findings, together with results from the quantitative counterpart of the RCT showing enhanced mindfulness and self-compassion, suggest that a range of qualities were experienced and cultivated. These were physical, psychological, social and spiritual, of nature and were experienced to support self-regulation, during and after a five-day nature-based mindfulness intervention.

## Figures and Tables

**Figure 1 healthcare-11-00905-f001:**
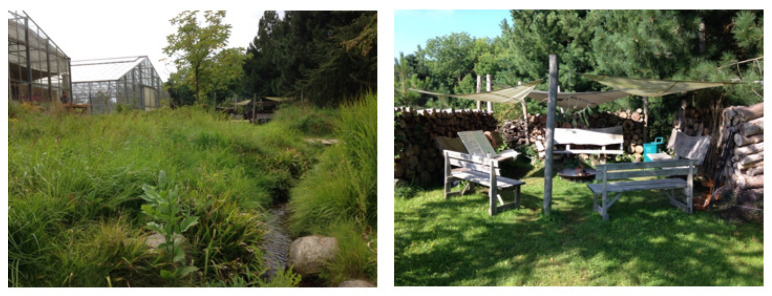
The Therapy Garden Nacadia^®^. Photos: Trine Fryjana Theede.

**Figure 2 healthcare-11-00905-f002:**
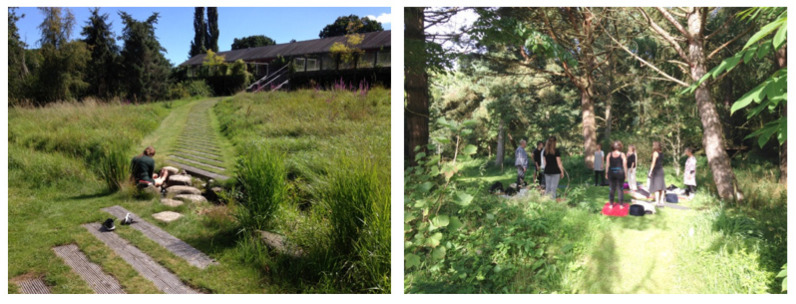
The participants in The Therapy Garden Nacadia^®^ Photos: Trine Fryjana Theede.

**Figure 3 healthcare-11-00905-f003:**
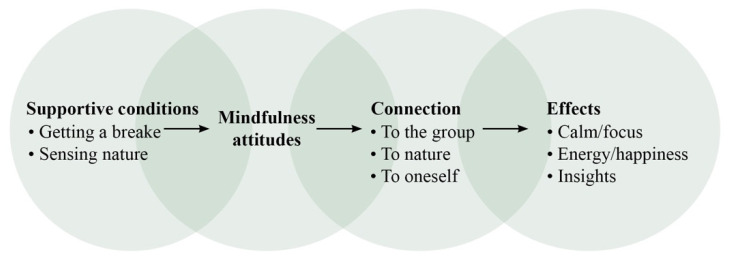
All themes were experienced to various extent during most of the retreat and perceived as supportive for self-regulation. The figure though depicts the general progression during the retreat.

## Data Availability

Data can be obtained from ddj@fondenmentalsundhed.dk.

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
