# Peer review of "Nature-Based Mindfulness: A Qualitative Study of the Experience of Support for Self-Regulation"

_healthcare, 2023, doi:10.3390/healthcare11060905_

Round 1

Reviewer 1 Report

Thank you for the opportunity to review the manuscript entitled, “Nature-Based Mindfulness: A Qualitative Study of the Experience of Support for Self-Regulation.”

 Below I provide my overall impressions followed by more specific comments.

OVERALL COMMENTS:

1.     This manuscript draws attention to an important topic in the effectiveness of mindfulness intervention and self-regulation in young adults.

2.     Parts of this manuscript could be strengthened. For example, including more details on the rationale to select NBMI for the current study. Why choose NBMI among other mindfulness intervention programs? Also, mention the exclusion criteria.

3.     Please provide a more detailed literature review regarding mindfulness and self-regulation.

4.     Please provide more detailed information regarding the sample selection process and exclusion criteria, if any.

Reviewer 2 Report

Thank you for your paper.

The topic of relationship between mental health and connection with nature is of growing preoccupation.

The introduction part is good and well written, please complete with information about IPA.

Your refer to a quantitative analysis too, but nothing is detailed about it.

We have no information as to what the results of the other 2 sub-groups were, here a quantitative analysis would help.

Please explain why the interviews were done in 2 different modes, on-site and online, and what the impact on the results could be due to this procedure.

Please explain the practical implications of your findings.
